# The impact of urinary incontinence on falls: A systematic review and meta-analysis

Shinje Moon[1], Hye Soo Chung[1], Yoon Jung Kim[1], Sung Jin Kim[2], Ohseong Kwon[2], Young Goo Lee[2], Jae Myung Yu[1], Sung Tae Cho[2]*

1 Division of Endocrinology and Metabolism, Department of Internal Medicine, Hallym University Kangnam Sacred Heart Hospital, Hallym University College of Medicine, Seoul, South Korea, 2 Department of Urology, Hallym University Kangnam Sacred Heart Hospital, Hallym University College of Medicine, Seoul, South Korea

* cst326@paran.com

**Data Availability Statement:** The literature search was conducted in adherence to the principles outlined by the Preferred Reporting Items for Systematic Reviews and Meta-Analyses.

## Abstract

### Objective

Previous studies on the association between urinary incontinence (UI) and falls have reported conflicting results. We, therefore, aimed to evaluate and clarify this association through a systematic review and meta-analysis of relevant studies.

### Methods

We performed a literature search for relevant studies in databases including PubMed and EMBASE from inception up to December 13, 2020, using several search terms related to UI and falls. Based on the data reported in these studies, we calculated the pooled odds ratios (ORs) for falls and the corresponding 95% confidence intervals (CIs) using the Mantel–Haenszel method.

### Results

This meta-analysis included 38 articles and a total of 230,129 participants. UI was significantly associated with falls (OR, 1.62; 95% CI, 1.45–1.83). Subgroup analyses based on the age and sex of the participants revealed a significant association between UI and falls in older (≥65 years) participants (OR, 1.59; 95% CI, 1.31–1.93), and in both men (OR, 1.88; 95% CI, 1.57–2.25) and women (OR, 1.41; 95% CI, 1.29–1.54). Subgroup analysis based on the definition of falls revealed a significant association between UI and falls (≥1 fall event) (OR, 1.61; 95% CI, 1.42–1.82) and recurrent falls (≥2 fall events) (OR, 1.63; 95% CI, 1.49–1.78). According to the UI type, a significant association between UI and falls was observed in patients with urgency UI (OR, 1.76; 95% CI, 1.15–1.70) and those with stress UI (OR, 1.73; 95% CI, 1.39–2.15).

### Conclusions

This meta-analysis, which was based on evidence from a review of the published literature, clearly demonstrated that UI is an important risk factor for falls in both general and older populations.

**Funding:** The authors declare that they have no relevant financial interests. (no funding).

**Competing interests:** The authors declare no conflicts of interest.

**Abbreviations:** CI, confidence interval; LUTS, lower urinary tract symptoms; OR, odds ratio; UI, urinary incontinence; WHO, World Health Organization.

## Introduction

The proportion of adults aged ≥65 years is increasing more rapidly than that of people in other age groups because of the global increase in life expectancy. However, this increase in life expectancy also increases the risk of geriatric syndromes, which are defined as the set of multifactorial conditions affecting older adults who are vulnerable to the changing circumstances [1]. Inouye et al. reported a high prevalence of five geriatric syndromes, namely, falls, incontinence, pressure ulcers, delirium, and functional decline, which are associated with high morbidity and poor quality of life [1].

Of these geriatric syndromes, falls represent one of the most important and increasing public health problems affecting older adults because these events often require medical attention. The World Health Organization (WHO) defines falls as "events that result in a person coming to rest inadvertently on the ground or floor or other lower-level." These events are often recurrent, and approximately half of the affected individuals experience another fall within 1 year [2]. According to the WHO, 28–35% of people older than 65 years of age fall each year, and this prevalence increases with age [3]. Another study determined that more than 30% of older (>65 years) home-dwelling individuals fall at least once per year [4]. Consequently, a substantial proportion of these individuals develop serious injuries, pain, depression, and other comorbidities. Even a slight fall can cause a fracture, which increases the risk of institutionalization and the associated economic burden. Falls also instill a source of fear in caregivers and negatively affect the healthcare systems [3]. In summary, falls result in negative health outcomes and limit the quality of life of older individuals, and strategies to prevent this geriatric syndrome should be established.

Assessing the association between falls and other geriatric syndromes [1] is clinically important in preventing falls. This syndrome is highly prevalent in the general population and affects men and women of all ages. Of the other geriatric syndromes, urinary incontinence (UI) is more common in women than in men; however, and the prevalence increases with age. Current estimates suggest that approximately 20 million women and 6 million men in the United States experience UI during their lives. This condition has been shown to affect 11–34% of men and 13–50% of women older than 60 years and 43–80% of all older nursing home residents [5]. UI is associated with not only a decreased quality of life but also a longer hospital stay and a reduced chance of hospital discharge [5]. However, many patients, particularly older individuals, avoid or do not receive treatment for UI due to the social stigma attached to the condition.

Although several epidemiological studies have evaluated the effects of UI on falls, the results of analyses based on age, sex, and the definition of falls have been inconclusive. Although some studies reported that UI is positively associated with falls [6–8], others indicated no association [9–11]. Hence, a meta-analysis was warranted to clarify our understanding of the role of UI in falls. We, therefore, performed a meta-analysis to provide evidence and determine the effect of UI on the risk of falls based on a comprehensive investigation of the literature. Furthermore, we conducted subgroup analyses based on patients' mean age, sex, the definition of falls, and type of UI.

## Methods

### Search strategy

A literature search was conducted in adherence to the principles outlined by the Preferred Reporting Items for Systematic Reviews and Meta-Analyses—PRISMA (S1 Table). The study protocol was registered in PROSPERO (CRD42021225038). Two independent investigators

(S.M. and S.T.C.) searched citation databases (PubMed, EMBASE, and Web of Science) for relevant studies. The search terms were a combination of "urinary incontinence" and "fall." The search was limited to original articles written in English and published between database inception and December 13, 2020 (S2 Table).

## Study selection

The inclusion criteria were as follows: 1) population: studies with participants aged $\geq$ 50 years or mean age $\geq$ 60 years; 2) exposure: the presence of UI; 3) comparators: participants without UI; 4) outcomes: incidence of falls; and 5) study design: case-control or cohort studies.

The exclusion criteria were as follows: 1) articles published as experimental studies, containing only abstracts, and published as non-original articles, including expert opinions or reviews; 2) studies that enrolled young adults aged <40 years; 3) observational studies without a control group.

## Data extraction

Data of the following variables were extracted independently by two investigators using the same criteria: name of the first author, year of publication, country, demographic characteristics of the participants, mean age of the participants, number of study participants, number of cases of falls, and odds ratios (OR) with 95% confidence intervals (CI).

## Risk of bias assessment

We used the Risk Of Bias In Non-randomized Studies—of Exposures (ROBINS-E), a modified form of ROBINS—of Interventions (ROBINS-I), to assess the methodological quality of the included studies [12, 13]. Discrepancies were resolved by discussion with a third investigator (J.M.Y).

## Data analyses and statistical methods

The overall ORs and 95% CIs of all studies were computed using the Mantel–Haenszel method. Heterogeneity among the studies was tested using the Higgins $I^2$ statistic, where an $I^2$ of $\geq$50% indicated heterogeneity. We computed the ORs using the random-effects model. Publication bias was calculated using a funnel plot and Egger's test. Sensitivity analysis was also performed.

## Subgroup analysis

All analyses were conducted using the Comprehensive Meta-Analysis software version 3 (Biostat, Englewood, NJ, USA).

## Results

### Study characteristics

In total, 1,427 studies were identified from the literature search (PubMed: 286, EMBASE: 439, and Web of Science: 702). After excluding 250 duplicate studies, we reviewed the remaining studies. Next, 1,177 studies were excluded during primary screening. After reviewing the texts of 107 articles, we excluded 73 studies, resulting in the inclusion of 34 articles [2, 7–11, 14–40]. In addition, we found four eligible studies from a previous review [41]. Finally, a total of 38 studies with 230,129 participants were included in this meta-analysis (Fig 1).

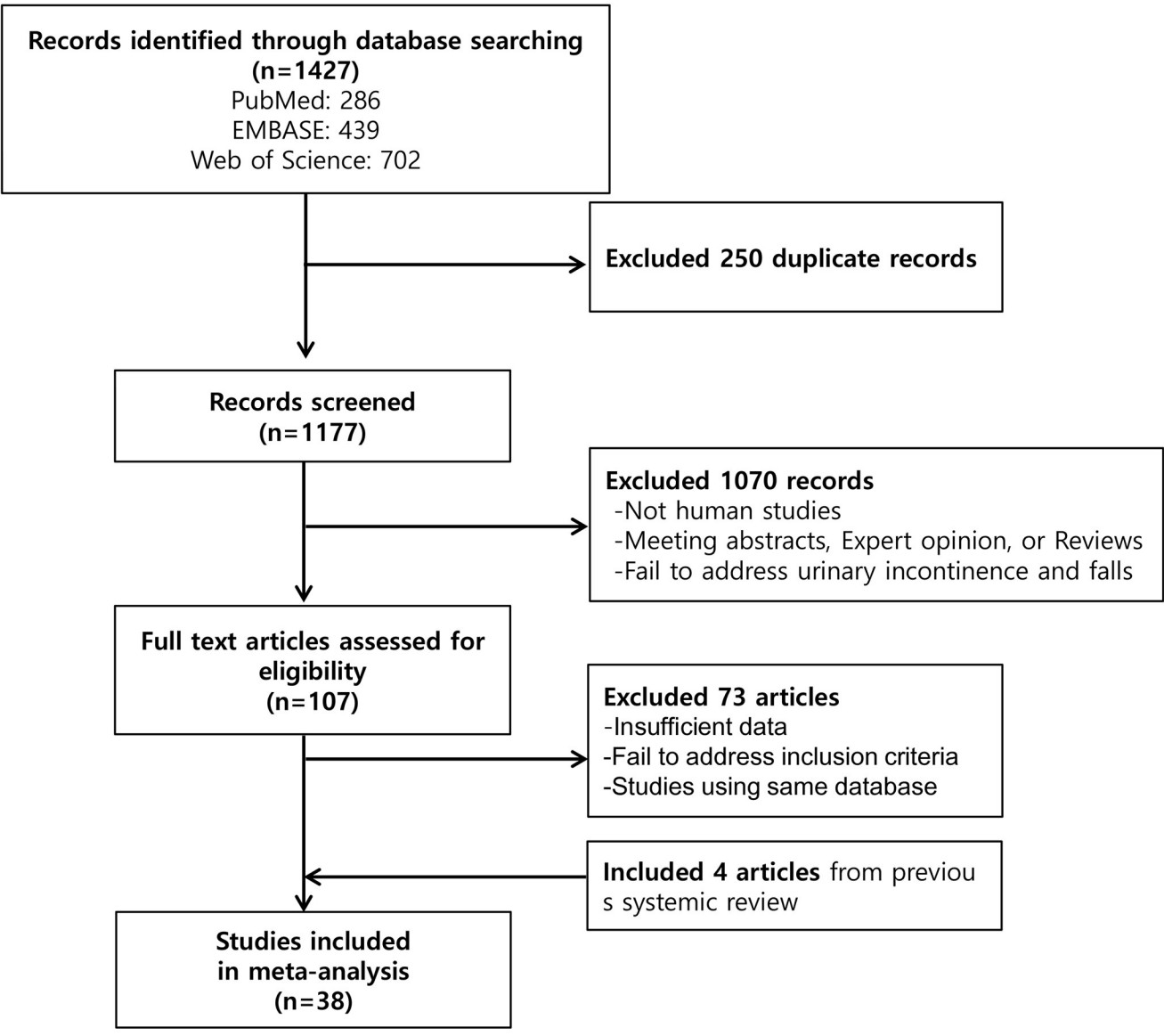

**Fig 1. Schematic diagram of the search strategy.**

The main characteristics of the studies are summarized in Table 1 [2, 7–11, 14–40, 42–46]. The meta-analysis revealed that, overall, 27.6% of participants (n = 63,618) experienced falls. The definitions of falls varied across the reviewed studies. Twenty-nine studies defined a fall as ≥1 fall event [7–9, 11, 17, 18, 20, 22, 23, 26–40, 42–46], four studies defined a fall as ≥2 fall events [14, 15, 21, 25], and five studies defined a fall as ≥1 fall event and recurrent falls as ≥2 fall events [2, 10, 16, 19, 24].

Fig 2 summarizes the quality assessment results of the studies and shows that the major source of bias in the studies bias was the lack of adjustment for potential confounders. Among the 38 studies, 14 studies did not adjust for confounding factors and were classified as studies with a critical risk of bias [8–11, 14, 20, 22, 25–27, 30, 39, 45, 46]. Seventeen studies had a serious risk of bias since more than one critically important confounding factor, namely age, sex, and physical function, was not appropriately adjusted or UI was not properly defined [15–19,

**Table 1. Summary of the 38 studies included in the present meta-analysis.**

| Study [Reference] | Country | Source of sample | Population characteristics | No. of total participants | Definition of falls/ No. of participants with falls | Definition and type of UI/ No. of participants with UI | Relative risk (95% CI) | Risk of bias |
|---|---|---|---|---|---|---|---|---|
| Tinetti ME et al. 1995 [14] | USA | Community dwelling adults, aged 72 years and older | Mean age: 79.7 Women:73% | 927 | At least two falls in 1 year | At least one UI / week in 1 year | Crude OR: 1.9 (95% CI, 1.2–2.9) | Critical |
| | | | | | 96 | 146 | | |
| Luukinen H et al. 1996 [15] | Finland | Community dwelling adults, aged 70 years and older | Mean age: 76.1 | 1,016 | At least two falls in 1 year | UI during the past 2 years | Adjusted [a] OR: 1.70 (95% CI, 1.03–2.89) | Serious |
| | | | Men: 396 Women: 620 | | 88 | 158 | | |
| Johansson C et al. 1996 [9] | Sweden | Community dwelling women, aged 85 year old | Mean age: 85 | 658 | At least one falls | UI monthly, weekly, several/week, daily, several/day. | Crude OR: 1.00 (95% CI, 0.75–1.33) | Critical |
| | | | Women:100% | | 286 | urge type (46%), | | |
| | | | | | | stress (21%), mixed (33%) | | |
| | | | | | | 384 | | |
| Brown JS et al. 2000 [7] | USA | Community-dwelling women, aged 65 years and older | Mean age: 78.5 | 6,049 | At least one falls in 1 year | UI during the past 1 year. | Adjusted [b] OR: | Moderate |
| | | | Women:100% | | 1,927 | At least one UI:2,818 (46.6%), | -Stress type: 1.06 (95% CI, 0.95–1.19) | |
| | | | | | | At least weekly urge type: 1,493 (24.7%), At least weekly stress type: 1,137 (18.8%), Both type: 708 (11.7%). | -Urge type: 1.26 (95% CI, 1.14–1.40) | |
| Tromp AM et al. 2001 [16] | Netherlands | Community-dwelling adults, aged 65 years and older | Mean age: 72.6 | 1,469 | At least one falls in 1 year | Self- reported UI 24% | Adjusted [c] OR: 1.6 (95% CI, 1.2–2.1) | Serious |
| | | | Men: 705 Women: 764 | | 464 | | Recurrent fall: Adjusted OR: 1.7 (95% CI, 1.2–2.5) | |
| de Rekeneire N et al. 2003 [17] | USA | Community-dwelling adults, aged 70 to 79 years | Age (70–79) Men: 1,447 | | At least one falls in 1 year | Self- reported UI1,175 | Adjusted [d] OR: | Serious |
| | | | Women: 1,515 | 2,962 | 652 | | - Men 1.5 (95% CI, 1.1–2.0) | |
| | | | | | | | - Women: 1.5 (95% CI, 1.2–1.9) | |
| Takazawa K et al. 2005 [10] | Japan | Women in a day care service at geriatric health facility | Median age: 81 | 118 | At least one falls in 1 year | At least once a week during the past 1 year | Crude OR: 1.12 (95% CI, 0.54–2.32) | Critical |
| | | | Women:100% | | 56 | Stress type: 25 (49.0%), Urge type: 46(90.2%) 52 | | |
| Teo JS et al. 2006 [18] | Australia | Community-dwelling women | Mean age: 79.1 | 782 | At least one falls in 1 year | Self- reported UI (regardless of amount and frequency) | Adjusted [e] OR: -Stress type: 1.06 (95% CI, 0.77–1.45) | Serious |
| | | | Women:100% | | 275 | | -Urge type: 1.96 (95% CI, 1.45–2.65) | |
| | | | | | | Stress type: 69.4% (pure 36.8%) | | |
| | | | | | | Urge type: 36.3% (pure 3.7%), both type: 32.6%. | | |
| | | | | | | 73.1% | | |
| Hasegawa J et al. 2010 [19] | Japan | Disabled older people who were admitted to facilities | Mean age: 82.5 | 1,082 | At least one falls | UI events during placement 180 | Adjusted [f] OR: 2.14 (95% CI, 1.03–2.89) | Serious |
| | | | Men: 327 | | 264 | | | |
| | | | Women: 755 | | | | | |
| Foley AL et al. 2012 [8] | UK | Community-dwelling adults aged 70 years or Over | Median age: 76 | 5,474 | At least one falls in 1 year | Self- reported UI | Crude OR: | Critical |
| | | | Men: 2,245 | | 1,813 | Stress type: 16.5%, urge type: 24.9% | -Stress type: 3.56 (95% CI, 3.06–4.15) | |
| | | | Women: 2,917 | | 26.7% | -Urge type: 2.19 (95% CI, 1.92–2.49) | | |

(*Continued*)

**Table 1.** (Continued)

| Study [Reference] | Country | Source of sample | Population characteristics | No. of total participants | Definition of falls/ No. of participants with falls | Definition and type of UI/ No. of participants with UI | Relative risk (95% CI) | Risk of bias |
|---|---|---|---|---|---|---|---|---|
| Allain TJ et al. 2014 [20] | Malawi | Community-dwelling adults aged 60 years or Over | Mean age: 72 | 98 | At least one falls in 1 year | Self- reported UI | Crude OR: 3.27 (95% CI, 1.26–8.50) | Critical |
| | | | Men: 29 | | | | | |
| | | | Women: 69 | | 40 | 25% | | |
| Huang LK et al. 2015 [21] | Taiwan | Community-dwelling adults aged 65 years or Over | Age ≥65 years | 187 | At least two falls in 1 year | UI in the past 1 year and 1week. | Adjusted [g] OR: 1.86 (95% CI, 0.86–4.02) | Serious |
| | | | Men: 65 | | | | | |
| | | | Women: 122 | | 53 | 29.9% | | |
| Kim H et al. 2015 [22] | Japan | Community-dwelling women aged 75–84 years | Mean age: 78.5 | 1,399 | At least one falls | UI over once a week | Crude OR: 1.57 (95% CI, 1.14–2.16) | Critical |
| | | | Women:100% | | 269 | Stress type: 29.2% (76/260), Urge type: 25.0% (65/260), and Mixed type:45.8% (119/260) | | |
| | | | | | | 260 | | |
| Sakushima K et al. 2016 [11] | Japan | Ambulatory patients with Parkinson's disease in an outpatient clinic of an academic hospital | Mean age: 71.5 | 97 | At least one falls in 6 months | Mild: less than once a day, severe: once a day or more past 1 week. | Crude OR: 2.05 (95% CI, 0.88–4.73) | Critical |
| | | | Men: 40 | | 44 | | | |
| | | | Women: 57 | | | Mild 27 | | |
| | | | | | | Severe 17 | | |
| Schluter PJ et al. 2018 [23] | New Zealand | Community-dwelling adults aged 65 years or Over | Mean age: 82.7 | 67,288 | At least one falls in 90 days | UI in the last 3 days | Adjusted [h] OR: | Moderate |
| | | | | | | Occasional UI: less than daily, frequently UI: daily | -Men | |
| | | | Men: 25,257 | | 27,213 | | | |
| | | | Women: 42,032 | | | Men 34.3% | Occasional UI 1.53 (95% CI, 1.43–1.64) | |
| | | | | | | Women 42.6% | | |
| | | | | | | | Frequent UI 1.69 (95% CI, 1.57–1.82) | |
| | | | | | | | -Women | |
| | | | | | | | Occasional UI | |
| | | | | | | | 1.33 (95% CI, 1.26–1.39) | |
| | | | | | | | Frequent UI 1.39 (95% CI, 1.32–1.46) | |
| Agudelo-Botero M et al. 2018 [24] | Mexico | Community-dwelling adults aged 60 years or Over | Age ≥60 years | 9,598 | At least one falls in 2 years | UI during the last 2 years | Adjusted [i] OR: -Occasional falls | Moderate |
| | | | Men: 4,271 | | | | | |
| | | | Women: 5,327 | | 4,466 (46%, one fall 16%, recurrent falls 30%)) | 3,021 | 1.12 (95% CI, 0.98–1.28) | |
| | | | | | | | -Recurrent falls 1.52 (95% CI, 1.37–1.69) | |
| Kang J et al. 2018 [25] | Korea | Patients older than 65 who visited the geriatric clinic | Mean age: 73 | 404 | At least two falls in 6 months | UI during the last 1 month | Crude OR: 2.07 (95% CI, 1.23–3.35) | Critical |
| | | | Men: 114 | | | | | |
| | | | Women: 290 | | 89 | 133 | | |
| Kim HJ et al. 2018 [26] | Korea | Community-dwelling adults aged 66 years or over in nationwide cohort study | Age (66–80) Men: 20,943 | 39,854 | At least one falls in 6 months | Self- reported UI | Crude OR: 5.29 (95% CI, 4.87–5.73) | Critical |
| | | | Women: 18,911 | | 2,802 | 5,703 | | |
| Sohn K et al. 2018 [27] | Korea | Community-dwelling women aged 65 years or over in Korean Longitudinal Study of Ageing | Age ≥65 years | 2,418 | At least one falls in 2 years | UI in the past 1 year | Crude OR: 1.29 (95% CI, 0.92–1.79) | Critical |
| | | | Women:100% | | 204 | 506 | | |
| Singh DKA et al. 2019 [28] | Malaysia | Community-dwelling adults aged 60 years or Over | Mean age: 68.9 | 3,901 | At least one falls in 1 year | Self- reported UI | Adjusted [j] OR: 1.35 (95% CI, 1.07–1.69) | Serious |
| | | | Men: 1,807 Women: 2,127 | | 804 | 615 | | |

(*Continued*)

**Table 1.** (*Continued*)

| Study [Reference] | Country | Source of sample | Population characteristics | No. of total participants | Definition of falls/ No. of participants with falls | Definition and type of UI/ No. of participants with UI | Relative risk (95% CI) | Risk of bias |
|---|---|---|---|---|---|---|---|---|
| Peeters G et al. 2019 [29] | Australia, Netherlands, Great Britain Ireland | Community-dwelling adults from four cohort (ALSWH, LASA, NSHD, TILDA) | Mean age: -ALSWH: 55.0–63.1. | ALSWH: 10,641 | At least one falls in 1 year | Self- reported UI -ALSWH: 45.6–59.0% | Adjusted [a] OR: -ALSWH: | Serious |
| | | | | LASA: 802 | -ALSWH: 2,352 | | | |
| | | | | | -LASA: 201 | -LASA: 16.7% | 1.53 (95% CI, 1.44–1.63) -LASA: | |
| | | | Women:100%-LASA: 59.7 | NSHD: 2,987 | -NSHD: 520 | -NSHD: 32.2% | 1.62 (95% CI, 0.95–2.78) | |
| | | | Women:51.6% | TILDA: 4663 | -TILDA: 820 | -TILDA: 10.3–12.6% | | |
| | | | -NSHD: 53.5–63.4 | | | | -NSHD: 1.68 (95% CI, | |
| | | | Women:50.9–52.2% | | | | 1.22–2.31) -TILDA: 2.09 (95% CI, 1.75–2.49) | |
| | | | -TILDA: 56.7–58.6 | | | | | |
| | | | Women:55.5–57.3 | | | | | |
| Giraldo-Rodriguez L et al. 2019 [30] | Mexico | Community-dwelling adults aged 50 years or Over | Aged ≥ 50 | 13,626 | At least one falls in 2 years | UI during the past 2 years | Crude OR: - Men: 1.42 | Critical |
| | | | Men: 5,843 | | 5,341 | -Men: 730 (12.5%) | (95% CI, 1.18–1.71) | |
| | | | Women: 7,783 | | | Stress type:141(2.4%), urge type:317(5.4%), mixed type:272(4.7%) | | |
| | | | | | | -Women: 2,155 | - Women: | |
| | | | | | | (27.7%) | 1.22 (95% CI, 1.06–1.39) | |
| | | | | | | Stress type:731(9.4%), urge type:488(6.3%), mixed type:936(12%) | | |
| Huang MH et al. 2019 [31] | USA | Men aged 65 years or over who had prostate cancer or breast cancer | 74.5(men) | 1097 | At least one falls in 1 years | UI during the past 6 months | - Men Adjusted [k] | Serious |
| | | | 75.1(women) | | | | | |
| | | | Men: 660 | | 231 | 285(men) | OR: | |
| | | | Women: 437 | | | 219 (women) | 1.69 (95% CI, 1.08–2.65) | |
| | | | | | | | -Women | |
| | | | | | | | Crude OR: 2.27 (0.89–5.80) | |
| Abbs E et al. 2020 [32] | USA | Homeless adults aged 50 years or Over | Median age: 58 | 350 | At least one falls in the past 6 months | UI during the past 6 months 167 | Adjusted [l] OR: 1.40 (95% CI, 1.07–1.81) | Moderate |
| | | | Men: 270 | | | | | |
| | | | Women: 80 | | 118 | | | |
| Abell JG et al. 2020 [33] | UK | Community-dwelling adults aged 60 years or Over | Mean age: 69.6 | 3,783 | At least one falls in 1 year | UI during the past 12 months | Adjusted [m] HR:: 1.49 (95% CI, 1.14–1.95) | Moderate |
| | | | Men: 1,791 | | 315 | 574 | | |
| | | | Women: 1,992 | | | | | |
| Britting S et al. 2020 [34] | Austria | Community-dwelling adults aged 75 years or over from SCOPE cohort | Median age: 79.5 | 2,256 | At least one falls in 1 year | UI during the last 1 month | Adjusted [n] OR: 1.33 (95% CI, 1.09–1.63) | Moderate |
| | Germany | | Men: 1,000 | | 746 | 653 | | |
| | Israel | | Women: 1,256 | | | | | |
| | Italy | | | | | | | |
| | Netherlands | | | | | | | |
| | Poland | | | | | | | |
| | Spain | | | | | | | |

(*Continued*)

**Table 1.** (Continued)

| Study [Reference] | Country | Source of sample | Population characteristics | No. of total participants | Definition of falls/ No. of participants with falls | Definition and type of UI/ No. of participants with UI | Relative risk (95% CI) | Risk of bias |
|---|---|---|---|---|---|---|---|---|
| Dokuzlar O et al. 2020 a [35] | Turkey | Women aged 65 years or over | Mean age: 74.4 | 682 | At least one falls in 1 year | UI during the past 12 months | Adjusted [o] OR: 1.61 (p value: 0.006) | Serious |
| | | | Women:100% | | 215 | 55.4% | | |
| Dokuzlar O et al. 2020 b [36] | Turkey | Men aged 65 years or over | Mean age: 75.0 | 334 | At least one falls in 1 year | UI during the past 12 months | Adjusted [o] OR: 2.468 (p-value: 0.001) | Serious |
| | | | Men:100% | | 85 | 33.2% | | |
| Lee K et al. 2020 [37] | USA | Community-dwelling adults aged 65 years or over | Mean age: 70.4 | 17,712 | At least one falls in 2 year | UI during the past 12 months | Adjusted [p] OR: 1.96 (95% CI, 1.59–2.40) | Serious |
| | | | Men: 7,626 | | 4,779 | 3,340 | | |
| | | | Women: 10,086 | | | | | |
| Magnuszewski L et al. 2020 [38] | Poland | Patients admitted to the department of geriatrics | Mean age: 85 | 358 | At least one falls in 1 year | Self- reported UI | Adjusted [q] OR: 1.37 (95% CI, 0.75–2.49) | Serious |
| | | | Men:80 | | | 146 | | |
| | | | Women: 278 | | 157 | | | |
| Moon S et al. 2020 [2] | Korea | Community-dwelling women aged 65 years or over | Mean age: 74.5 | 6,134 | At least one falls in 1 year | Self- reported UI | Adjusted [r] OR: 1.33 (95% CI, 1.00–1.76) | Moderate |
| | | | Women:100% | | 1,152 | 281 | | |
| Savas S et al. 2020 [39] | Turkey | Community-dwelling adult | Mean age: 65 | 1176 | At least one falls in 1 year | Self- reported UI | Crude OR: | Critical |
| | | | Men:592 | | | 346 | 1.21 (95% CI, 0.79–1.87) | |
| | | | Women: 584 | | 276 | | | |
| Tsai YJ et al. 2020 [40] | Taiwan | Community-dwelling adults aged 65 years or over (NHIS 2005, 2009, 2013) | Men:4,142 | 8,822 | At least one falls in 1 year | Self- reported UI | Adjusted [s] OR: 1.09 (0.80–1.49), 1.29 (0.90–1.84), 1.42 (1.04–1.94) | Serious |
| | | | Women: 4,680 | | 1,672 | 1,573 | | |
| Cesari M et al.2002 [42] | Italy | Community-dwelling adults admitted to national home care program | Mean age: 77.2 | 5,570 | At least one falls in 90 days | Self- reported UI | Adjusted [t] OR: 1.06 (0.93–1.20), | Serious |
| | | | | | 1,997 | 1,744 | | |
| | | | | Men: 2,290 | | | | |
| | | | | Women: 3,280 | | | | |
| Hedman AM et al. 2013 [43] | Sweden | Community-dwelling adults aged 75 years or over | Median age: 81 | 1,243 | At least one falls in 1 year | Self- reported UI 1,139 | Adjusted [u] OR: 1.53 (1.23–1.91), | Serious |
| | | | Men: 471 | | 434 | 425(men) | | |
| | | | Women: 772 | | | 714(women) | - Men: 1.67 (1.13–2.47), | |
| | | | | | | | - Women: 1.53 (1.16–2.00) | |
| Moreira MD et al. 2007 [46] | Brazil | Community-dwelling adults aged 60 years or over | Mean age: 79 | 490 | At least one falls in 1 year | Self- reported UI 86 | p <0,025 | Critical |
| | | | Men: 116 | | | | | |
| | | | Women: 374 | | 137 | | | |
| Stenhagen M et al. 2013 [44] | Sweden | Community-dwelling adults aged 60 years or over | Men: 264 | 1,736 | At least one falls in 6 months | Self- reported UI | Crude OR: 1.89 (1.38–2.58) | Serious |
| | | | Women: 394 (3-year follow up) | 555 (3-year follow up) | 106(3-year follow up) | 267 | | |
| | | | Men: 784 | 1,542(6-year follow up) | 205(6-year follow up) | | With UI: 267 (67 with falls) | |
| | | | Women: 963 (6-year follow up) | | | | Without UI: 1453 (219 with falls) | |
| | | | | | | | Adjusted [a] OR: 1.31 (0.94–1.82) | |

(*Continued*)

**Table 1.** (Continued)

| Study [Reference] | Country | Source of sample | Population characteristics | No. of total participants | Definition of falls/ No. of participants with falls | Definition and type of UI/ No. of participants with UI | Relative risk (95% CI) | Risk of bias |
|---|---|---|---|---|---|---|---|---|
| van Helden S et al. 2007 [45] | Netherland | Patients older than 50 who visited the geriatric clinic | Mean age: 67.1 | 277 | At least one falls in 3 months | Self- reported UI | Crude OR: 2.07 (0.98–4.41) | Critical |
| | | | Men: 77 | | 42 | 50 | | |
| | | | Women: 200 | | | | | |

OR, odds ratios; HR, hazard ratios; UI, urinary incontinence; CI, confidence intervals; ALSWH, The Australian Longitudinal Study on Women's Health; LASA, The Longitudinal Ageing Study Amsterdam; NSHD, The MRC National Survey of Health and Development; TILDA, The Irish Longitudinal Study on Ageing.

[a] adjusted for age and sex,

[b] adjusted for age, living situation, overall frailty, number of falls in the previous year, whether she walked for exercise, alcohol and caffeine consumption, medical history, medication use, grip strength, gait speed, whether she used her arms to stand from chair, and performance of 10-second tandem balance.

[c] adjusted for age, gender, educational level, urbanization level, chronic diseases, physical function, level of activity and mobility, previous falls, fear of falling

[d] adjusted for age, race, study site, and body mass index.

[e] adjusted for age, central nervous system drug and cardiovascular system drugs.

[f] adjusted for age, gender, physical function, behavioral symptom, and medication use.

[g] adjusted for gender, depressive mood, and activities involving lower limb.

[h] adjusted for: age, ethnicity, marital status, living arrangements, body mass index, cognitive performance, dementia, congestive heart failure, Chronic obstructive pulmonary disease, depression, diabetes mellitus, alcohol consumption, smoking status, hearing status, vision status, fatigue, mobility, stability, dizziness, wandering, season, bisphosphonates, vitamin D, and calcium.

[i] adjusted for sociodemographic, medical and functional covariables.

[j] adjusted for age, sex, educational level and ethnicity.

[k] adjusted for age at prostate cancer diagnosis, time since cancer diagnosis, history of falls, marital status, physical summary score of Veterans RAND 12-Item Health Survey.

[l] adjusted for age, sex, race, stroke, Activities of Daily Living (ADL) impairment, use of an assistive device., marijuana use, opioid use, history of physical assault, any nights spent in unsheltered settings.

[m] adjusted for age, sex, chronic conditions (coronary heart disease, diabetes, Stroke, Arthritis, Osteoporosis, Parkinson's Disease), BMI, Smoking status, Alcohol consumption, The Short Physical Performance Battery (SPPB), and history of severe fall.

[n] adjusted for age, gender, geriatric depression score (GDS), chronic kidney disease (CKD), instrumental activities of daily living (IADL) score and Euro-Qol 5D Score.

[o] adjusted for age, education level, and living environment.

[p] adjusted for age, sex, race/ethnicity, and spouse/partner status.

[q] adjusted for age, multimorbidity, chronic diseases (cardiac heart failure, peripheral arterial disease, history of stroke/ transient ischemic attack, Parkinson's disease, and chronic osteoarthritis, Performance Oriented Mobility Assessment, Barthel Index, IADL score, gait speed, Clinical Frailty Scale, Mini Nutritional Assessment Short Form, albumin value, vitamin B12 level and taking certain medications (quetiapine, vitamin D, diuretics, benzodiazepines and selective serotonin reuptake inhibitor).

[r] adjusted for age, smoking status, alcohol consumption, body mass index, hypertension, dyslipidemia, and diabetes mellitus, cognitive impairment, ADL and IADL disability, visual and hearing impairment, and lower limb weakness.

[s] adjusted for age, sex, developing difficulty in performing ADLs or IADLs, use of sleeping pills, vision, comorbidities, depressive symptoms, and frequency of exercise.

[t] adjusted for age, gender, activities of daily living impairment, foot problems, gait problems, fear of falling, visual impairment, wandering, depression, parkinsonism, and environmental hazards.

[u] Poor self-rated health, Pain in neck and shoulders, Back pain, sciatica or hip pain, Pain in hands, elbows, legs or knees, Headache or migraine, Anxiety, Tiredness, Sleeping disorders, Tinnitus, Recurring stomach problems, Overweight/Underweight.

21, 28, 29, 31, 35–38, 40, 42–44]. Seven studies, which were appropriately adjusted for confounding factors, had a moderate risk of bias [2, 7, 23, 24, 32–34].

## Impact of UI on falls

According to the random-effects model, the overall OR for falls was 1.62 (95% CI, 1.45–1.83). An overall $I^2$ of 96.0% indicated heterogeneity among the studies (Fig 3). The funnel plot and

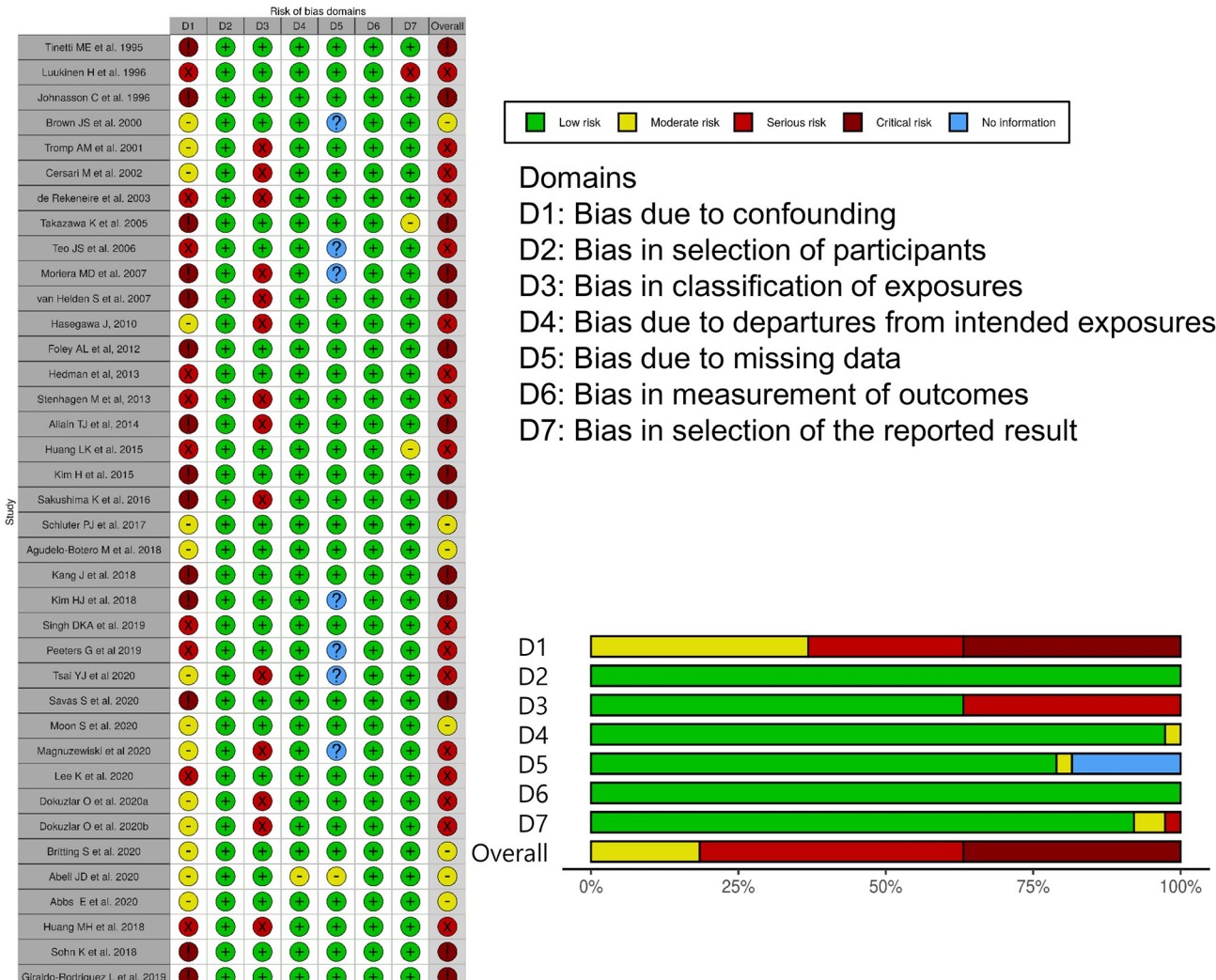

**Fig 2. Quality assessment of the risk of bias in the 33 studies included in this meta-analysis.**

Egger's test did not reveal any publication bias (p = 0.477, Fig 4A). The sensitivity analysis revealed consistently significant ORs between 1.55 and 1.67, even after excluding the results of each included study (Fig 4B). After excluding 14 studies with a critical risk of bias, the OR was 1.46 (95% CI, 1.38–1.56; $I^2$, 76.5%).

## Analyses of subgroups stratified by age, sex, the definition of falls, and type of UI

Subgroup analyses were performed according to the age and sex of the participants (Table 2). A significant association between UI and falls was observed in older adults (≥65 years; OR, 1.59; 95% CI, 1.31–1.93) [2, 7–10, 14–17, 21–23, 25–27, 31, 34–37, 40, 43], and in both men (OR, 1.88; 95% CI, 1.57–2.25) [17, 23, 29–31, 36] and women (OR, 1.41; 95% CI, 1.29–1.54) [2, 7, 9, 10, 17, 18, 22, 23, 27, 29–31, 35, 45]. In a subgroup analysis of 34 studies that defined falls as ≥1 fall event, the OR for the association between UI and falls was 1.61 (95% CI, 1.42–1.82; $I^2$, 96.3%; Table 2) [2, 7–11, 16–20, 22–24, 26–40, 42–46]. In a subgroup analysis of nine

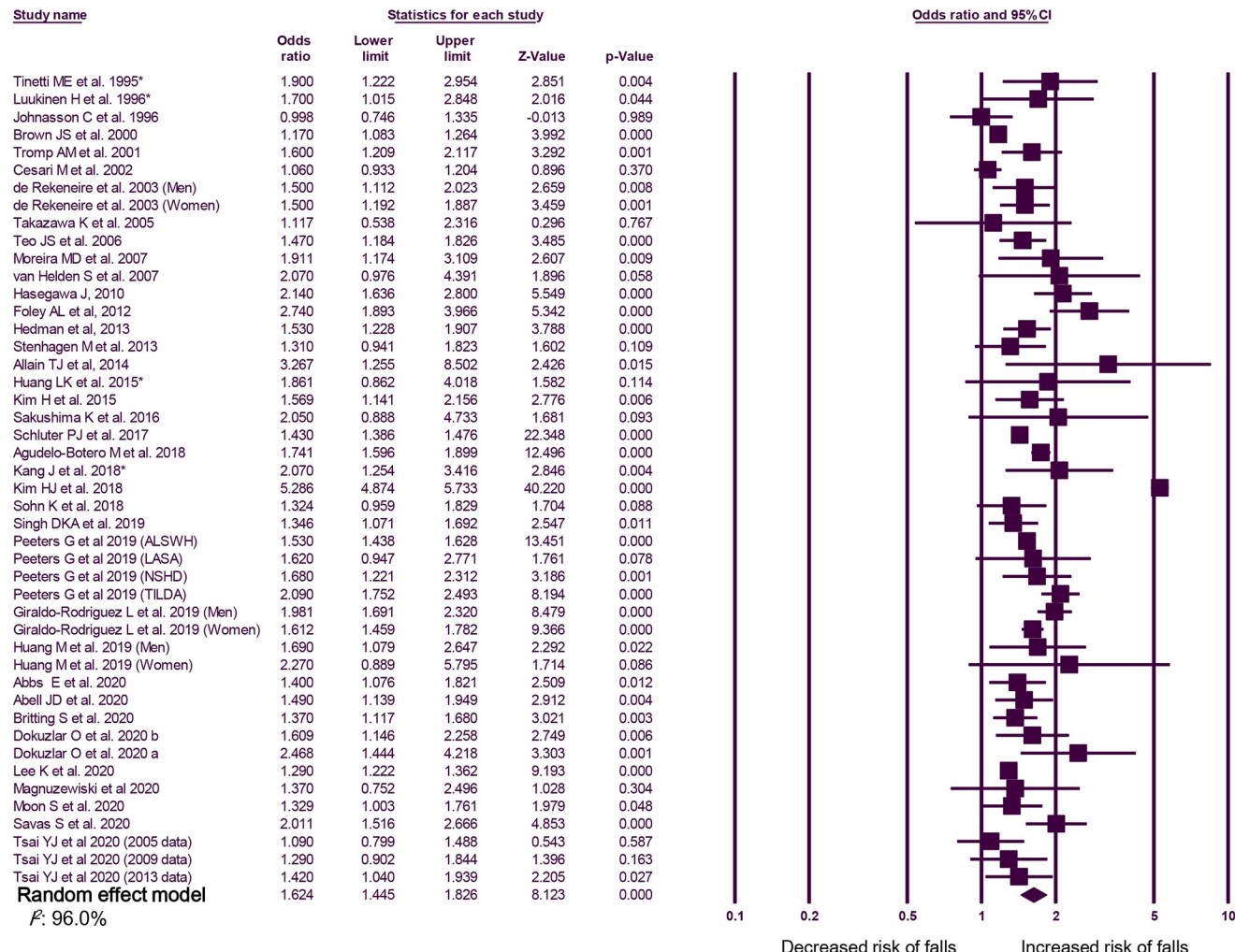

**Fig 3. Forest plots of the risk ratio of the association between urinary incontinence and falls.** OR, odds ratio; CI, confidence interval. *Study that defined falls as at least two falls within 1 year.

studies that defined recurrent falls as $\geq$2 fall events, the OR for the association between UI and falls was 1.63 (95% CI, 1.49–1.78; $I^2$, 40.6%; Table 2) [2, 10, 14–16, 19, 21, 24, 25]. In a subgroup analysis according to the type of UI, a significant association between UI and falls was observed in patients with urgency UI (OR, 1.76; 95% CI, 1.15–1.70) [7, 8, 10, 18, 30] and in those with stress UI (OR, 1.73; 95% CI, 1.39–2.15) [7, 8, 10, 18, 30].

## Discussion

Although UI is a known risk factor for falls, the strength of the association between these conditions remains unclear because of variability in the study designs and populations used in previous risk estimations. This systematic review and meta-analysis conducted to evaluate the association between falls and UI revealed that UI was associated with overall falls. Our analysis identified a probable excess OR of 65% for at least one fall among people with UI relative to those without UI. An analysis of participants with recurrent falls yielded a similar trend and a higher risk magnitude. The overall OR for recurrent falls was 63% among people with UI relative to those without UI.

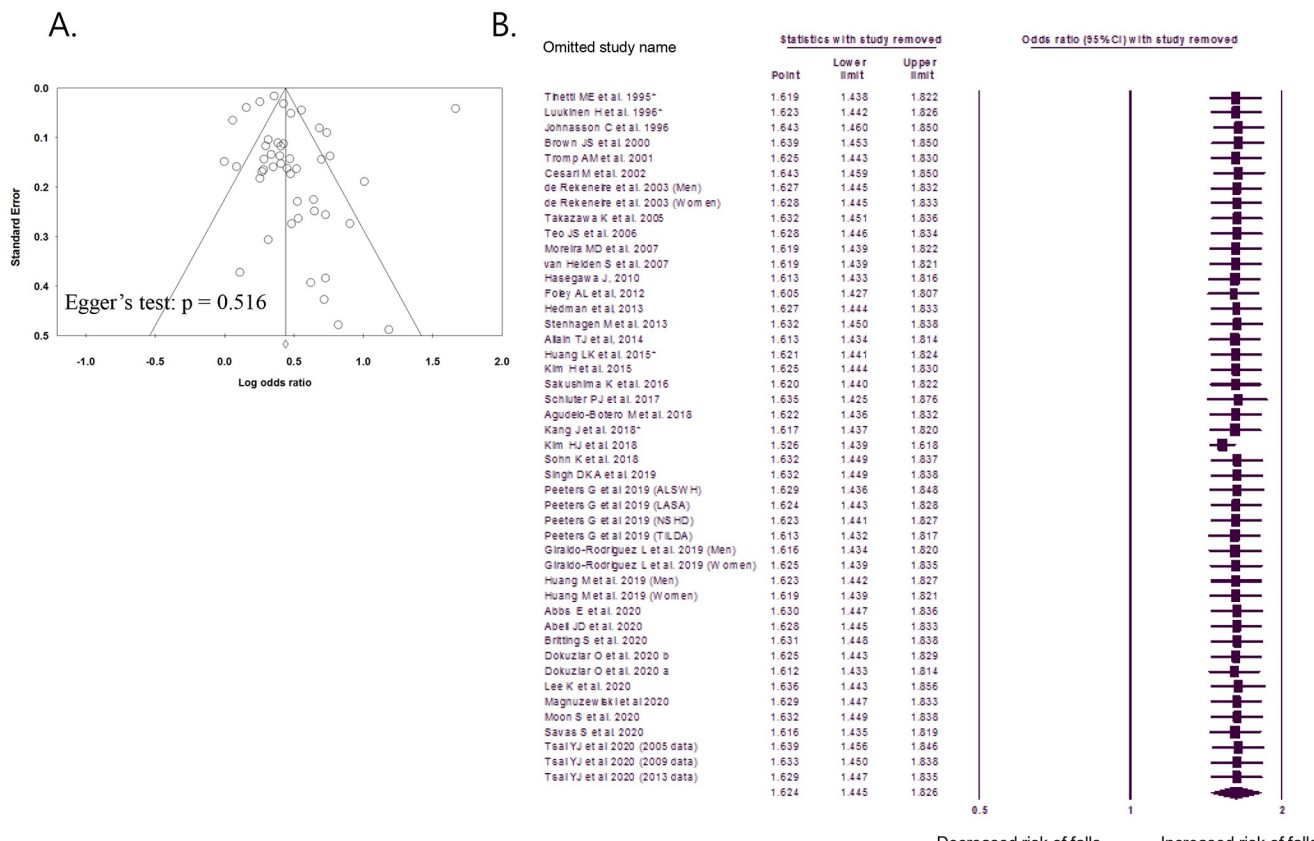

**Fig 4. Funnel plot and sensitivity analysis.** A. Funnel plot of publication bias in studies comparing the odds ratios of urinary incontinence for falls. B. Sensitivity analysis of the meta-analysis of studies comparing the odds ratios of urinary incontinence for falls. *Study that defined falls as at least two falls within 1 year.

In a subgroup analysis, we determined that the OR for falls increased by 59% in older adults (≥65 years) with UI relative to those without UI. These findings exceed those of older systematic reviews that considered a more limited range of fall-related outcomes and consistently reported an increased risk of falls and fractures among participants with UI [47]. UI is of significant concern to older adults and can lead to isolation and reduced self-worth. Previous

**Table 2. Subgroup analysis of the association between urinary incontinence and falls.**

| Subgroup | No. of studies [Reference] | OR (95% CI) | Heterogeneity ($I^2$), % |
|---|---|---|---|
| Age, ≥ 65 years | 22 [2, 7–10, 14–17, 21–23, 25–27, 31, 34–37, 40, 43] | 1.59 (1.31–1.93) | 97.6% |
| Sex | | | |
| Men | 6 [17, 23, 29–31, 36] | 1.88 (1.57–2.25) | 75.2% |
| Women | 14 [2, 7, 9, 10, 17, 18, 22, 23, 27, 29–31, 35, 45] | 1.41 (1.29–1.54) | 79.5% |
| Definition of falls | | | |
| Falls ≥ 1 | 34 [2, 7–11, 16–20, 22–24, 26–40, 42–46] | 1.61 (1.42–1.82) | 96.3% |
| Falls ≥ 2 | 9 [2, 10, 14–16, 19, 21, 24, 25] | 1.63 (1.49–1.78) | 40.6% |
| Type of urinary incontinence | | | |
| Urgency incontinence | 5 [7, 8, 10, 18, 30] | 1.76 (1.15–1.70) | 97.1% |
| Stress incontinence | 5 [7, 8, 10, 18, 30] | 1.73 (1.39–2.15) | 90.2% |

studies have identified various risk factors for falls, such as old age, female sex, visual disturbances, cognitive disorders, low body mass index, and UI.

We conducted another subgroup analysis according to the type of UI. A previous review highlighted a predominant association of falls with urgency UI, rather than with other types of UI [47]. This association is attributed to the urgent need to use the toilet and the anxiety associated with a failure to reach the toilet. Several studies have shown that behavioral changes induced by UI can affect the likelihood of falls [48, 49]. Our analysis also showed a higher risk of falls in patients with urgency UI than in those with stress UI. Falls related to this condition have been generally reported to occur in the toilet [7, 47]. Despite this relationship, however, the commonly held assumption that urgency leads to falls while rushing to the toilet has not been confirmed yet [6].

Few studies have investigated the relationship between UI and falls [47], and the causality between UI and falls remains unexplained [6]. However, one hypothesis is that a strong desire to void could change gait parameters and thus, increase the risk of falls [50]. The reduced velocity and stride width during strong desire to void conditions (i.e., urgency) in the UI group could explain their high fall rate [50]. The other hypothesis is that women with impaired mobility probably take a longer time to reach the toilet; hence, if there is a high degree of urgency, then impaired mobility can increase the risk of UI [51]. Therefore, the causality between UI and falls could probably be explained by a strong desire to void and physical impairments in mobility and balance [50, 51]. However, although these hypotheses could explain the relationship between the urgency-type UI and falls, they are rather insufficient to explain the association between stress-type UI and falls. Since the symptoms of urgency UI and stress UI are clinically different, the association between stress UI and falls may indicate a general alteration in the striated muscle physiology in the aging population [8]. In addition, restricted mobility in older women may limit their ability to change positions to prevent stress UI [22].

There is a well-recognized association between falls and lower urinary tract symptoms (LUTS) in older adults [7, 8, 47, 52, 53]. Older people with urgency or urgency UI are significantly more likely to fall than age-matched controls, with ORs for falls ranging between 1.5 and 2.3 [6, 47, 54, 55]. However, the reason for this association is not understood and has not been thoroughly studied [6].

In a recent systemic review on the association between falls and LUTS conducted by Noguchi et al., none of the identified studies had investigated the potential causes of these associations. In addition, the categorization of UI and degree of accounting for confounding variables were inconsistent across the studies [56]. Although the data identified were suitable only for qualitative synthesis, UI and storage symptoms among LUTS have been consistently reported to have a weak to moderate association with falls [6, 56].

As our findings suggest that this association is significant, the identification and treatment of UI may be an effective intervention for reducing the risk of falls, especially in older adults. Bladder training, timed or prompt voiding, and environmental modifications (e.g., a bedside commode) may decrease the incidence of falls [7].

Concerning the impact of UI on the risk of falling, many falls are related to a person's physical condition or medical problems, such as multimorbidities, polypharmacy, neurological diseases, and sarcopenia, as well as urological comorbidities [57]. Especially, multiple medications, such as blood pressure-lowering drugs causing orthostatic hypotension, psychotropics, anticonvulsants, and sedatives, can contribute to falls [57]. In addition, the geriatric syndrome has a multifactorial etiology, with the factors being closely related to each other [1]. Among them, UI and falls are very important for the older population, and both are associated with sarcopenia [8, 58, 59]. Therefore, an appropriate statistical approach to decrease the impact of

such confounding variables is necessary for correct analysis of the association between UI and falls.

The strengths of this study include the collection of evidence through a rigorous systematic review and meta-analysis. This study also included a comprehensive search of both published and unpublished studies. Multiple measurements of falls were considered, consistent with multiple types of risk estimates. Although many studies have included UI as a risk factor for falls, only a few studies have identified UI as an individual risk factor [47]. Therefore, this is the first systematic review and meta-analysis to evaluate UI as an individual risk factor for falls.

Despite these strengths, our study was limited largely by the included studies, particularly the significant heterogeneity, quality of the study designs, and reporting scope of the original articles. However, when studies with a critical risk of bias were excluded, significant results were observed. In addition, no publication bias was observed, and the results were not changed by specific studies in the sensitivity analysis. Furthermore, although we conducted subgroup analyses based on age, sex, and type of UI, we did not perform analyses according to the severity of UI. Finally, the paucity of evidence regarding the severity of UI limits the applicability of our current findings with regard to an accurate correlation between UI and falls.

In conclusion, the continued increase in the proportion of older adults globally will lead to continued increases in the clinical and economic impacts of serious falls. Based on evidence from the published literature and a meta-analysis, we demonstrate here that UI is a predictor of more frequent falls in both general and older adults. Clinicians should, therefore, be aware that UI predicts an increased risk of falls that could lead to fractures and should, therefore, provide appropriate precautions and care. Future studies are needed to address the impact of UI treatment on the incidence of falls.

## Supporting information

**S1 Table. PRISMA checklist.**
(DOCX)

**S2 Table. Electronic search strategy.**
(DOCX)

**S1 Data. PubMed: 286 studies.**
(DOCX)

## Author Contributions

**Conceptualization:** Shinje Moon, Sung Tae Cho.

**Data curation:** Shinje Moon.

**Formal analysis:** Shinje Moon.

**Investigation:** Shinje Moon, Sung Jin Kim, Ohseong Kwon, Young Goo Lee, Jae Myung Yu.

**Methodology:** Shinje Moon, Hye Soo Chung, Yoon Jung Kim, Sung Tae Cho.

**Project administration:** Sung Tae Cho.

**Supervision:** Sung Tae Cho.

**Validation:** Sung Tae Cho.

**Visualization:** Sung Tae Cho.

**Writing – original draft:** Shinje Moon, Sung Tae Cho.

**Writing – review & editing:** Shinje Moon, Sung Tae Cho.

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
