## [Decision Letter · Decision Letter 0]

5 Nov 2020

PONE-D-20-28464

The Impact of Urinary Incontinence on Falls: A Systematic Review and Meta-Analysis

PLOS ONE

Dear Dr. Cho,

Thank you for submitting your manuscript to PLOS ONE. After careful consideration, we feel that it has merit but does not fully meet PLOS ONE’s publication criteria as it currently stands. Therefore, we invite you to submit a

ACADEMIC EDITOR:

In addition to the reviewers' comments, I have a comment. The populations in the studies you have selected are quite different. Some studies have included elderly people with Parkinson's disease or multiple sclerosis, and others elderly people diagnosed with frailty. These people with impaired mobility and / or strength might have a different starting risk if incontinence (or frequent micturition) is present, or develops than in the cohorts without these disorders. Doesn't this require a separate paragraph in the discussion on how this (confounder?) is handled in the manuscripts? You also mention the term 'prompt voiding', the standard term for this strategy is 'prompted voiding', I think.

We look forward to receiving your revised manuscript.

Kind regards,

Peter F.W.M. Rosier, M.D. PhD

Academic Editor

PLOS ONE

Journal Requirements:

Additional Editor Comments (if provided):

Reviewers' comments:

Reviewer's Responses to Questions

**Comments to the Author**

1. Is the manuscript technically sound, and do the data support the conclusions?

Reviewer #1: Partly

Reviewer #2: Yes

2. Has the statistical analysis been performed appropriately and rigorously? 

Reviewer #1: I Don't Know

Reviewer #2: Yes

3. Have the authors made all data underlying the findings in their manuscript fully available?

Reviewer #1: No

Reviewer #2: Yes

4. Is the manuscript presented in an intelligible fashion and written in standard English?

Reviewer #1: Yes

Reviewer #2: Yes

5. Review Comments to the Author

Reviewer #1: In this systematic review with meta-analysis, the authors focused on the relationship between urinary incontinence and falls. Although the subject is fascinating, the study is penalized by an incomplete submission (tables 1 to 3 are missing) and the absence of subgroup analysis concerning the type of urinary incontinence (urgency vs. stress). It is therefore possible to improve this manuscript by taking into account these 2 points and those that follow:

General comment: Authors should avoid the term "elderly" or "elder" and prefer "older"

1. Abstract: There is a difference between the abstract and the methods in the dates of literature search (october 2019 in the abstract and april 2020 in the methods)

2. Methods:

a. Why did the authors choose as an eligibility criteria studies with population over 20 years? The introduction is developed around the ageing of the population and the fact that both urinary incontinence and falls are geriatric syndromes. I do know that the definition of older age varies widely among countries but 20 years old is definitely not old…

b. Results of the study selection process should be included in the results and not in the methods

c. Data extraction: there is no information on urinary incontinence, its type or severity.

d. Quality assessment. This section should be renamed as: risk of bias assessment. Unfortunately, there is no validation study of the Newcastle Ottawa tool, and no validation of the cut-off used to assess the risk of bias (what is the reference for this cut-off?) Could the authors discuss the choice of this tool and use a more appropriate one?

e. Why the authors did not register their systematic review and meta-analysis protocol? It is well known that the absence of a published protocol can lead to bias (10.1001/jama.287.21.2831).

f. Regarding the search strategy, there is no full electronic search strategy available as it is stated in the PRISMA checklist. Authors should provide it as a supplementary data.

3. Results:

a. The results section is uncomplete since tables 1 to 3 are missing (problem during the submission process?). It is difficult to judge this work without these tables, and I cannot make much more comments without these tables.

b. It seems that there is a problem with the figure 3 (“omitting” in front of each study name). Did the authors proofread their manuscript?

c. The study by Peeters et al in 2019 was included 4 times in the meta-analysis. Could the authors provide an explanation? Are they sure that it was not conducted in the same population?

4. Discussion:

Since we do not have any tables, it is difficult to understand why the authors did not discuss a critical point. I think that the association between UI and falls is rather known and admitted, but the remaining question is “is there any causality between UI and falls”? As the authors said, UI and falls are probably more related to “frailty” and the significant association that many studies highlight may not exist as such.

o Do we have any prospective data to answer this question?

o Do we have some biological hypothesis to support this? (rather not, the authors started to discuss this point but not enough to my opinion, they forgot to include in there discussion the studies published by Dumoulin et al. on gait and urge to void, by Fritel et al. on urinary incontinence subtypes and mobility impairment, etc. )

o Were the studies adjusted on potential confounders? (neurological pathologies? Comorbidities? Polypharmacy?)

o Was the association different if stress or urgency urinary incontinence were considered?

Reviewer #2: I want to congratulate the authors for the idea and development of this manuscript. The paper adds valuable information about the link of falls and UI in terms of higher level of evidence after this systematic review.

I suggest some comments in order to improve quality and understanding of the manuscript:

1. Figure 4 can be omitted, from my view it does not add important information for the reader

2. Please describe acronyms TE/seTE in figure 2.

3. There is a significative statistical heterogeneity in the studies included. This is reflected in the results section, but it is a limitation of the study and need to be commented in the discussion. Probably this point and the quality of the study designs and the reporting scope of the original articles is the main limitation of this meta-analysis.

4. Some studies have shown association between type of UI (specially UUI) and severity (more number of UUI episodes per week). The authors don’t clarify why they don’t include this subgroup analysis. It would be very interesting to have this information in this kind of systematic review.

5. There is a previous systematic review on LUTS and falls that the authors has not mentioned [Noguchi, N., et al., A systematic review of the association between lower urinary tract symptoms and falls, injuries, and fractures in community-dwelling older men. Aging Male, 2016: p. 1-7.]. In this SR none of the identified studies examined potential causes for these associations; the categorisation of continence or not and degree of accounting for confounding variables was inconsistent across the included studies. This point should be discussed and compared with results with your SR.

6. From the point of view of the reader is not clear the categorization in the definition of falls (≥1, ≥2, ≥2 recurrent falls). It would be very helpful to provide a definition of falls (within which time lapse?, what is the definition of recurrent falls?).

6. PLOS authors have the option to publish the peer review history of their article (what does this mean?). If published, this will include your full peer review and any attached files.

Reviewer #1: **Yes: **Rebecca Haddad

Reviewer #2: No

---

## [Author Response · Author response to Decision Letter 0]

1 Feb 2021

PLOS ONE - Decision on Manuscript PONE-D-20-28464

Peter F.W.M. Rosier, M.D. PhD

Academic Editor

PLOS ONE

Dear Dr. Rosier:

On behalf of all of the authors for this manuscript, we really appreciate your meticulous comments as they refine this manuscript. Regarding these comments, the reviewer’s suggestions were implemented in the revised manuscript. 

The changes are summarized below in red.

Authors tried to revise the manuscript following their opinions as far as we can. We hope the revised manuscript will better meet the requirements of your journal for publication. We thank the editor and the reviews of PLOS ONE once again for the constructive review of our paper.

Sincerely yours,

Sung Tae Cho, MD, Ph.D.

Professor, Dept. of Urology,

Kangnam Sacred Heart Hospital

Hallym University College of Medicine

---

## [Decision Letter · Decision Letter 1]

11 Mar 2021

PONE-D-20-28464R1

The Impact of Urinary Incontinence on Falls: A Systematic Review and Meta-Analysis

PLOS ONE

Dear Dr. Cho,

Thank you for submitting your manuscript to PLOS ONE. After careful consideration, we feel that it has merit but does not fully meet PLOS ONE’s publication criteria as it currently stands. Therefore, we invite you to submit a revised version of the manuscript that addresses the points raised during the review process.

ACADEMIC EDITOR:

Your manuscript has been greatly improved, but, apart from the reviewers comments, I believe I am still missing the answer and / or the processing of my questions to the authors in my earlier letter. Therefore more or less again, in addition to the question of the reviewer to clarify the choice of age categories, I would also like to add some clarification regarding the studies that included persons with a disorder. This is to make it clearer how generalizable your conclusions can be, or how specific they should be. You may therefore have to check, in any case, whether your conclusion is not too general.

We look forward to receiving your revised manuscript.

Kind regards,

Peter F.W.M. Rosier, M.D. PhD

Academic Editor

PLOS ONE

Journal Requirements:

Reviewers' comments:

Reviewer's Responses to Questions

**Comments to the Author**

1. If the authors have adequately addressed your comments raised in a previous round of review and you feel that this manuscript is now acceptable for publication, you may indicate that here to bypass the “Comments to the Author” section, enter your conflict of interest statement in the “Confidential to Editor” section, and submit your "Accept" recommendation.

Reviewer #1: (No Response)

Reviewer #2: All comments have been addressed

2. Is the manuscript technically sound, and do the data support the conclusions?

Reviewer #1: Yes

Reviewer #2: Yes

3. Has the statistical analysis been performed appropriately and rigorously? 

Reviewer #1: Yes

Reviewer #2: Yes

4. Have the authors made all data underlying the findings in their manuscript fully available?

Reviewer #1: Yes

Reviewer #2: Yes

5. Is the manuscript presented in an intelligible fashion and written in standard English?

Reviewer #1: Yes

Reviewer #2: Yes

6. Review Comments to the Author

Reviewer #1: Thank you to the authors for taking our remarks into account and for modifying the manuscript accordingly, which has been considerably improved. Please find additional comments.

Methods

- It is still not clear why this age (≥40 years) was chosen in a study about an essentially geriatric problem. Authors must explain it. Why did they not choose ≥ 60 years old? I only found 2 studies in the results with a possible mean age under 60 (Giraldo Rodriguez L et al. 2019 [29] and Abbs E et al. 2020 [31]). This remark does not mean that I want the authors to remove these studies, but I really want to know the reason behind this selection criteria.

- Study selection: rather use “exposure” than “intervention”

Results

- In table 1, there are some missing informations:

* population characteristics: sex ratio (Tinetti, Johansson etc.), age (de Rekeneire, Huang, etc.) or both (Kim) are missing for some of the studies. Please fill in the missing informations.

* exposition: there is no information on the definition/type/prevalence of UI. Please fill in the missing informations.

* I would merge the 2 columns “definition of falls” and “no. of participants with falls” so that you can add a “UI” column.

* for adjusted OR, I would add, as a footnote under the table, the variables of adjustment.

* when OR is not available, please give the prevalence in the two groups.

* footnotes: missing explanation of the abbreviations: ALSWH, LASA, NSHD, TILDA, UI, CI

* For the study by Stenhagen et al., were the OR calculated on the 6 years or 3 years data? Please indicate the number of participants according to the OR calculation

- Risk of bias assessment

*Figure 2 risk of bias: The authors forgot to adapt the ROBINS-I template to the ROBINS-E tool. For example, D3 bias in classification of interventions is not applicable to the ROBINS-E tool and should be replace by “Bias in classification of exposures”. Please check every single item to be in accordance with the ROBINS-E tool.

*In the manuscript, I think you should explain a little bit more the risk of bias. At least write the major source of bias (bias due to confounding) and highlight the studies who correctly address confounding factors in the relationship between UI and falls.

- Figures 3 et 4 B

The “favours A/favours B” footnote does not work here since authors assess the effect of an exposure and not an intervention. Please modify to something more adequate to this analysis

Discussion

The authors have made some changes regarding the hypotheses to explain of the association between urge UI and falls. But the authors did not make any hypotheses to explain the association between stress UI and falls. To my opinion it is important, and it is one of the most interesting findings of this meta-analysis. Clinicians will remember that urinary incontinence of any type is a factor associated with falls. But authors must make some hypotheses. Is UI just a marker of poor health? Or is UI a real causal factor? Here authors should discuss a little bit more the studies that correctly address confounding factors, especially the ones that could lead to both UI and falls (for example multimorbidity, polypharmacy, neurological diseases, sarcopenia, etc.).

Reviewer #2: I have reviewed this new version of the manuscript. Reviewer's comments and questions have been fully addressed, and the manuscript has increased significantly its quality and understanding.

I am satisfied with the final result. No more comments to add.

7. PLOS authors have the option to publish the peer review history of their article (what does this mean?). If published, this will include your full peer review and any attached files.

Reviewer #1: **Yes: **Rébecca Haddad

Reviewer #2: **Yes: **Salvador Arlandis

---

## [Author Response · Author response to Decision Letter 1]

25 Apr 2021

PLOS ONE - Decision on Manuscript PONE-D-20-28464R1

Peter F.W.M. Rosier, M.D. PhD

Academic Editor

PLOS ONE

Dear Dr. Rosier:

We are pleased to re-submit our revised manuscript to the PLOS ONE. We express our sincere appreciation for your thoughtful comments. Following your suggestions, we describe answers about the issues raised by reviewers. In addition, we designate any changes by highlighting with red color in the revised manuscript.

Reviewers' comments:

Reviewer #1: Thank you to the authors for taking our remarks into account and for modifying the manuscript accordingly, which has been considerably improved. Please find additional comments.

Methods

- It is still not clear why this age (≥40 years) was chosen in a study about an essentially geriatric problem. Authors must explain it. Why did they not choose ≥ 60 years old? I only found 2 studies in the results with a possible mean age under 60 (Giraldo Rodriguez L et al. 2019 [29] and Abbs E et al. 2020 [31]). This remark does not mean that I want the authors to remove these studies, but I really want to know the reason behind this selection criteria.

Response: Thank you for careful comments. Considering the heterogeneous characteristics between young adults and older adults, we excluded the studies with young adults to remove their confounding effect on the results. However, considering several cohort studies often included middle aged people in their baseline survey, we thought that tight age criterion might miss large population based cohort studies such as Giraldo‐Rodríguez et al. Therefore, we chose the broad criterion (≥40 years) to screen as many studies as possible. In addition, we thought the heterogeneous effect of middle aged people could be solved by subgroup analysis stratified by age and conducted a subgroup analysis with studies in older adults (≥65 years). Nevertheless, we agreed the reviewer’ comment that this criterion could not explain enough of the aim of our study. Therefore, we modified this criterion as follow:

1) population: studies with participants aged ≥ 50 years or mean age ≥ 60 years

- Study selection: rather use “exposure” than “intervention”

Response: As reviewer’s comment, we revised the manuscript.

Results

- In table 1, there are some missing informations:

* population characteristics: sex ratio (Tinetti, Johansson etc.), age (de Rekeneire, Huang, etc.) or both (Kim) are missing for some of the studies. Please fill in the missing informations. 

* exposition: there is no information on the definition/type/prevalence of UI. Please fill in the missing informations.

* I would merge the 2 columns “definition of falls” and “no. of participants with falls” so that you can add a “UI” column.

* for adjusted OR, I would add, as a footnote under the table, the variables of adjustment.

* when OR is not available, please give the prevalence in the two groups.

* footnotes: missing explanation of the abbreviations: ALSWH, LASA, NSHD, TILDA, UI, CI

* For the study by Stenhagen et al., were the OR calculated on the 6 years or 3 years data? Please indicate the number of participants according to the OR calculation

Response: We agree with the reviewer. For all studies, we have checked again to find if there was any information missing. Then, we have added the missing data and modified the table 1. as the reviewer pointed out. 

- Risk of bias assessment

*Figure 2 risk of bias: The authors forgot to adapt the ROBINS-I template to the ROBINS-E tool. For example, D3 bias in classification of interventions is not applicable to the ROBINS-E tool and should be replace by “Bias in classification of exposures”. Please check every single item to be in accordance with the ROBINS-E tool.

Response: As reviewer’s comment, we revised figure 2. 

*In the manuscript, I think you should explain a little bit more the risk of bias. At least write the major source of bias (bias due to confounding) and highlight the studies who correctly address confounding factors in the relationship between UI and falls.

Thank you for careful comments. We explained more details about the risk of bias in manuscript. 

- Figures 3 et 4 B

The “favours A/favours B” footnote does not work here since authors assess the effect of an exposure and not an intervention. Please modify to something more adequate to this analysis

Response: As reviewer’s comment, we revised figure 3 and 4 as follow: 

favours A/favours B � Decreased risk of falls/Increased risk of falls

Discussion

The authors have made some changes regarding the hypotheses to explain of the association between urge UI and falls. But the authors did not make any hypotheses to explain the association between stress UI and falls. To my opinion it is important, and it is one of the most interesting findings of this meta-analysis. Clinicians will remember that UI of any type is a factor associated with falls. But authors must make some hypotheses. 

Response: We agree with the reviewer that the hypothesis of stress UI is different to those of urgency UI. Thus, we have added the sentence in the discussion section as the reviewer pointed out (page 24, lines 13-19).

Is UI just a marker of poor health? Or is UI a real causal factor? Here authors should discuss a little bit more the studies that correctly address confounding factors, especially the ones that could lead to both UI and falls (for example multimorbidity, polypharmacy, neurological diseases, sarcopenia, etc.).

Response: Thank you for your comments. Thus, we have performed the updated literature search and added the sentence about confounding factors such as multimorbidity, polypharmacy, neurological diseases and sarcopenia in the discussion section as the reviewer pointed out (page 25, lines 12-22).

We hope this revised work is impressive to your editor, reviewers, and readers. We thank the editor and the reviews of PLOS ONE once again for the warm review of our paper and look forward to hearing from you.

Sincerely yours,

Sung Tae Cho, MD, Ph.D.

Professor, Dept. of Urology,

Kangnam Sacred Heart Hospital

Hallym University College of Medicine

---

## [Editor Report · Decision Letter 2]

3 May 2021

The Impact of Urinary Incontinence on Falls: A Systematic Review and Meta-Analysis

PONE-D-20-28464R2

Dear Dr. Cho,

We’re pleased to inform you that your manuscript has been judged scientifically suitable for publication and will be formally accepted for publication once it meets all outstanding technical requirements.

Kind regards,

Peter F.W.M. Rosier, M.D. PhD

Academic Editor

PLOS ONE

Additional Editor Comments (optional):

none
---

## [Editor Report · Acceptance letter]

6 May 2021

PONE-D-20-28464R2 

The Impact of Urinary Incontinence on Falls: A Systematic Review and Meta-Analysis 

Dear Dr. Cho:

I'm pleased to inform you that your manuscript has been deemed suitable for publication in PLOS ONE. Congratulations! Your manuscript is now with our production department. 

Kind regards, 

on behalf of

Dr. Peter F.W.M. Rosier 

Academic Editor

PLOS ONE